# Exploring the perceptual straightness of adversarially robust and biologically-inspired visual representations

**Anne Harrington**[1,2]    **Vasha DuTell**[1,2]    **Ayush Tewari**[1]    **Mark Hamilton**[1,3]
**Simon Stent**[4]    **Ruth Rosenholtz**[1,2]    **William T. Freeman**[1]
[1]MIT CSAIL    [2]MIT Brain and Cognitive Sciences
[3] Microsoft Research    [4] Toyota Research Institute
{annekh,vasha}@mit.edu

## Abstract

Humans have been shown to use a "straightened" encoding to represent the natural visual world as it evolves in time (Hénaff et al. 2019). In the context of discrete video sequences, "straightened" means that changes between frames follow a more linear path in representation space at progressively deeper levels of processing. While deep convolutional networks are often proposed as models of human visual processing, many do not straighten natural videos. In this paper, we explore the relationship between robustness, biologically-inspired filtering mechanisms, and representational straightness in neural networks in response to time-varying input, and identify curvature as a useful way of evaluating neural network representations. We find that (1) adversarial training leads to straighter representations in both CNN and transformer-based architectures and (2) biologically-inspired elements increase straightness in the early stages of a network, but do not guarantee increased straightness in downstream layers of CNNs. Our results suggest that constraints like adversarial robustness bring computer vision models closer to human vision, but when incorporating biological mechanisms such as V1 filtering, additional modifications are needed to more fully align human and machine representations.

## 1   Intro

Visual input from the natural world evolves over time, and we can think of that change over time as a trajectory in some representation space. This trajectory changes at different levels of processing from input at the retina to brain regions such as V1, and finally to perception as figure 1 illustrates. We can ask about the shape of that trajectory and consider that there might be advantages to a straighter, less curved, trajectory. Hénaff et al. [18] observed that trajectories are straighter in human perceptual space than in pixel space. They suggested that a straighter representation may be useful for visual tasks that require predicting the future.

Learning a useful visual representation is also a goal in computer vision. Properties like robustness to transformations and task flexibility that characterize human vision are often desirable in computer vision representations. Yet, many existing computer vision models still fail to capture aspects of human vision, despite achieving high accuracy on visual tasks like recognition [15, 18]. In Hénaff et al. [18] it was found that, while biologically-inspired V1-like transformations yield straighter representations compared to the input domain, popular computer vision models such as ImageNet-trained AlexNet do not.

In this paper, we explore what makes some learned visual representations straighter than others. We evaluate how training for adversarial robustness in both CNN and transformer-based architectures may lead to the straight representations generated by human vision. Because DNNs learn an early

4th Workshop on Shared Visual Representations in Human and Machine Visual Intelligence (SVRHM) at the Neural Information Processing Systems (NeurIPS) conference 2022. New Orleans.

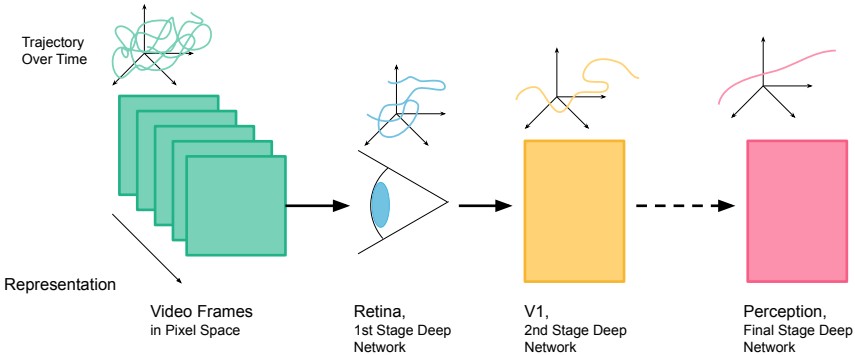

Figure 1: Illustration of the representation of a discrete video sequence becoming progressively straighter as information is processed from pixel space through a visual processing pipeline.

representation that differs from what's known about human vision, we also ask if hard-coding that early representation might lead to a trained network with more straightening. Overall, we find that deep learning models are capable of relying on a straightened representation which may be useful for learning more robust and stable image and video processing systems.

## 2   Previous Work

Deep Neural Networks (DNNs) have been proposed as models of human visual processing, owing to their ability to predict neural response patterns [33, 26, 21]. Comparing human and DNN perception [2, 15, 28] has shown that adversarial examples are an important area in understanding how humans and DNNs differ [12, 20, 16, 8]. With adversarial examples, changes that are imperceptible to humans can cause a network to misclassify an image. Adversarial training [23] improves the misclassfication problem and has been suggested to help networks learn visual representations that are more perceptually aligned with humans. [14, 20]. Given adversarial training schemes are not biologically plausible however, recent work has identified mechanisms that are better supported by vision science [7, 17, 8].

Adversarially robust models have also been shown to do better at transfer learning than their non-adversarially robust counterparts [9], and adversarially robust features can be used directly for tasks like image generation and in-painting [27]. The potential for adversarial training to improve learned representations has led to new adversarial training frameworks that extend to flow-based generative models [25] and semantic segmentation models [32]. In this paper, we build on work around adversarial robustness by evaluating if increasing this robustness leads to straightened representations like those found in human spatiotemporal processing.

## 3   Methods

Representational straightness can be evaluated as a reduction in curvature. For a temporal sequence, such as a video, curvature is defined as the angle between the vectors representing the *difference* between consecutive frames. Let $\mathbf{x}$ refer to a representation of a video of length $T$, with $x_t$ being a representation of one frame of a video at time step $t$. The representation may be at any stage of the processing pipeline, from a vector of raw input pixels from the video frame, to the activations of a network's hidden layer. Then $v_t$ represents the difference between successive frames:

$$v_t = x_t - x_{t-1} \tag{1}$$

$$\hat{v}_t = \frac{v_t}{\|v_t\|} \tag{2}$$

We can find the curvature at time $t$ by finding the angle between successive $\hat{v}_t$, which we call $c_t$:

$$c_t = \arccos\left(\hat{v}_t \cdot \hat{v}_{t+1}\right) \tag{3}$$

The global curvature of a video sequence is then simply the mean angle over all time steps:

$$\text{Global curvature} := \sum_{t=0}^{T-1} \frac{c_t}{T-1} \tag{4}$$

This is the formulation proposed by Hénaff et al. [18]. One can compute this global curvature for any representation of a video sequence over time, either on the vector of pixels (likely not very straight), or one can apply it to a representation of that video, e.g. at any layer of a neural network model.

## 4 Effect of Architecture and Training Scheme on Straightness

We tested a variety of models for output curvature, to investigate the relationship between model type and curvature of the output layer. As shown in Fig 2, we found non-adversarially trained image models to have the highest output curvature. All adversarially trained models have lower curvature than their non-adversarially trained counterparts, as well as overall. We also tested a family of non-parametric biological network models: PeriphNet [4], based on summary statistics of a steerable pyramid, and Henaffbio [18], a two-stage model based on center-surround filters followed by oriented Gabor filters. These biological models showed the lowest output curvatures. Self-supervised DINO [5] models have similar output curvature values to their supervised counterparts – despite DINO models having been shown to have more semantically meaningful feature correspondences. In addition, we investigated a next-frame prediction model, PredNet [22]. As PredNet is a predictive coding model for frames in pixel space, it produces output predictions with similar curvature to its video input.

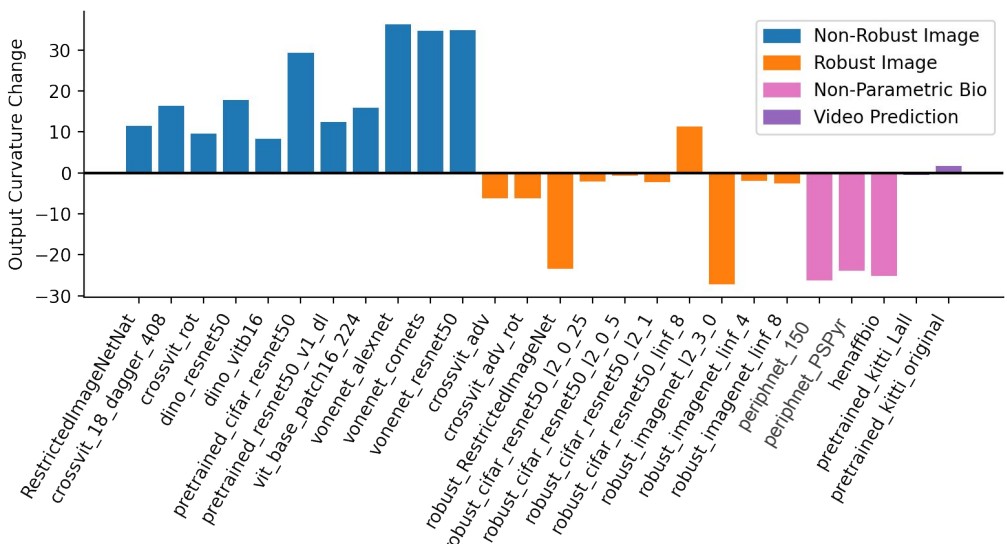

Figure 2: **Final output layer curvature for a variety of network architecture and training schemes.** Non-adversarially trained image models increase curvature. Most adversarially trained image models reduce curvature, resulting in a more straightened output. Non-parametric biological models, without learned filters, produce highly straightened output representations. Video models trained on next-frame prediction show output curvatures very similar to that of input pixels.

## 5 Adversarial Attacks in Image Models

Given the increased straightness seen in adversarially robust models, we investigated the relationship between the type and strength of adversarial attack and the resulting curvature of the model's output. To evaluate the effect of these attacks on curvature, we compared a set of ResNet50 networks, trained on CIFAR-10, ImageNet [10], and Restricted ImageNet (a subset of ImageNet [14, 20]), trained both without adversarial attacks, as well as trained on both $l_2$ and $l_\infty$ norms at a range of attack strengths [23, 13], and measured the output curvature of each resulting model (Figure 3). We find that

output curvature is unsurprisingly highest for non-adversarially trained networks. $l_\infty$ attacks decrease output curvature, with larger values of $\epsilon$ leading to decreased curvature. $l_2$ attacked networks have the straightest output curvatures, however strength of attack does not affect the output curvature.

Differences between the $l_2$ and $l_\infty$ norm may be the reason that we found $l_2$ norm training to lead to more straight representations. The $l_\infty$ norm takes the maximum entry of a vector whereas the $l_2$ norm is the square root of the sum of squares and is informed by the all components of a vector. Thus, the $l_2$ norm may allow updates in training to affect more weights and change the over the whole representation space to produce a more straight and stable representation over time.

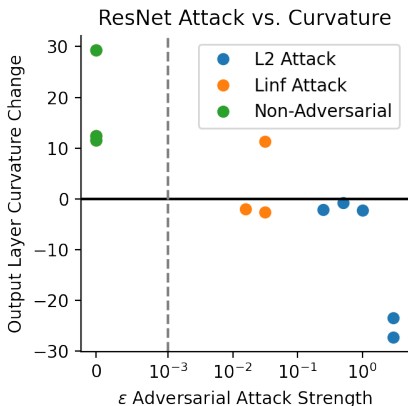

## 6 Curvature Across Model Layers

We next investigated the evolution of layer curvature of various deep network models over major model blocks from input to output (Figure 4). As in Figure 2, we again see that non-adversarially trained models increase curvature in the representation, while adversarially robust models show a straightened output. For most models, this trend is consistent throughout all layers of the network. For ResNet50 however, the adversarially trained model begins by curving the output in earlier layers, then reducing curvature strongly. Conversely, in the non-adversarially trained CrossVIT network, the curvature is reduced in middle layers, then highly curved in later layers.

Figure 3: White-box adversarial attacks reduce curvature in the output layer of ResNet. Increased attack strength ($\epsilon$) decreases curvature, with $l_2$ attacks most reducing curvature. Data on symlog x-scale to show $\epsilon = 0$: data to left of line on linear scale, and right of line on log scale.

We also tested the layer-evolution of curvature for PredNet (Fig 4), a network inspired by predictive coding trained to predict the next frame in a video sequence. Since PredNet's trained to predict a future video frame, it's input and output domains are both in pixel space, so it is unsurprising that the output layer maintains a very similar curvature value to that of its input frame. Interestingly however, the model's curvature strongly increases in the representation of the first model block, then re-straightens the representation throughout the rest of the network before returning to its starting curvature.

## 7 Biologically Inspired Models

We investigated straightness for a variety of both parametric and non-parametric biologically-inspired models. Given straightness is thought to increase over progressively deeper layers of visual processing, we aligned these networks along the visual processing areas they are most closely matched to (Figure

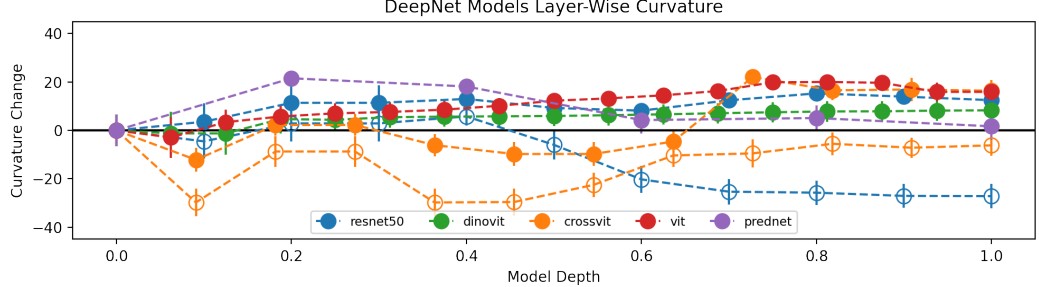

Figure 4: **Deep network models vary in curvature for each layer over model depth.** Curvature is shown for a variety of model types. Filled circles indicate non-adversarially trained models, and open circles indicate adversarially-trained robust models.

5). For the non-parametric Henaffbio [19] and PeriphNet [4] models, as well as VOneNetCornets [7] these are implicit in the design of the network. For the adversarially trained Visual Transformer network CrossVITRotAdv [3], these layers are those best matched by BrainScore for V4 and IT layers [28]. For all biologically-inspired models except for VOneNetCornets, curvature progressively decreases through deeper network layers. For VOneNetCornets, curvature decreases up until the V1 layer in which a noise term is added; curvature then strongly increases, far above the pixel-curvature baseline. To determine if this increase in curvature was due to the added noise, we tested the same model with the noise term set to zero. While this reduced the downstream curvature after the V1 layer, this change did not eliminate the curvature increase present in the V1 layer.

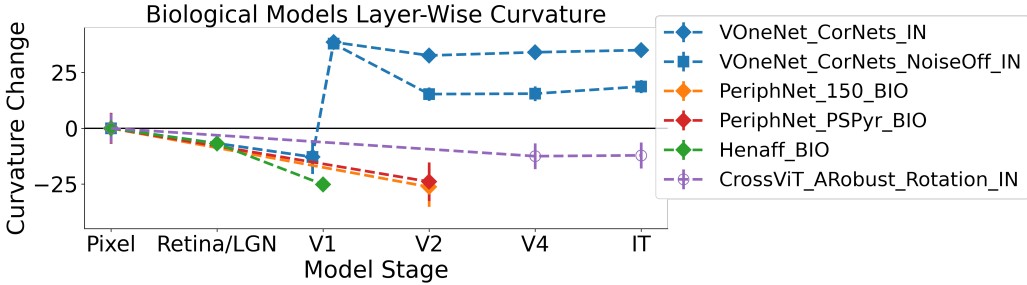

Figure 5: **Biologically-inspired network models result in straightened representations compared to input pixels.** Non-parametric multi-stage filter models PeriphNet and Henaffbio progressively straighten. The VOneNetCornets model straightens the representation in its V1-block up until its noise layer, where curvature increases dramatically and does not recover in downstream convolutional blocks. Adversarially-trained CrossVIT model shows progressive straightening.

The increase in curvature for VOneNet at later layers suggest that making the front-end of deep network more like biologically-inspired models is not enough to get a straightened representation downstream in a deep network. This is interesting because VOneNet is is reported to be more adversarially robust to white-box attacks than a standard trained ResNet. It suggests that adversarial *training*, not the property of adversarial robustness itself, leads to straightened representations in deep neural networks. Our finding supports Dapello et al. [8] who found that the neural population geometry of adversarial trained models was noticeable differences from VOneNet and other models trained with neural stochasticity mechanisms. However, it is puzzling that adversarial training, which is biologically implausible, would lead to more straight representations than biologically inspired mechanisms. More constraints or modifications may be need to get straight representations with biologically plausible methods.

## 8 Discussion

In conclusion, we show change in model representational curvature to be a simple and computationally cheap metric for evaluating both image and video models across a variety of tasks. We show that for a variety of image classification models, output curvature is reduced when models are trained with strong white-box adversarial attacks. This property of straightness over time may lead to more stable predictions over both input space and for temporal sequences. Although Hénaff et al. [18] found that ImageNet-trained DNNs did not yield perceptually straight representations when tested on videos, we find that this is not a limitation of the model but rather of the training procedure. Moreover, we find evidence to suggest that a model's ability to straighten input stimuli may be a useful and easily computed measure of its ability to produce similar visual representations to humans. In evaluating curvature over layers in biologically inspired models, we show that biologically inspired mechanisms work to reduce curvature in a model's representation, even more so than adversarial training. However, the simple addition of non-parametric biologically inspired filtering mechanisms at the input of a model are insufficient to maintain output curvature. These results identify representational curvature as a common thread between biologically inspired and adversarially robust models, and highlight the benefits and limitations of these techniques in creating temporally-stable representations.

## Acknowledgments

This work was funded by Toyota Research Institute and MIT Meteor Fellowship.

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

# A  Appendix

# B  Curvature Definition

Note that curvature is distinct from simple cosine similarity in that curvature is calculated on frame differences ($v_t$), whereas cosine similarity depends on the angle between the frame vectors themselves ($x_t$). Curvature can be thought of as a first order variant cosine similarity.

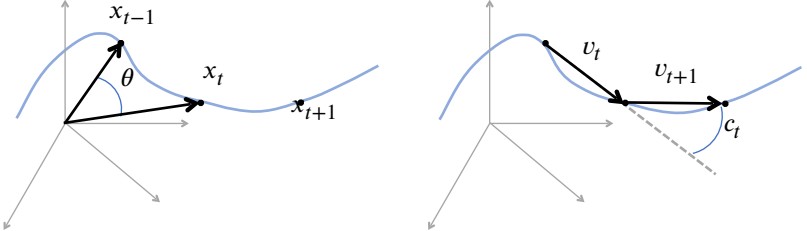

Figure 6: Illustration of how the curvature measure is distinct from cosine similarity. (Left) Three points are sampled along a trajectory in time ($x_{t-1}, x_t, x_{t+1}$). The angle $\theta$ between neighboring $x$ samples is their cosine similarity. (Right) Curvature $c_t$ is the angle between $v_1$ and $v_{t+1}$. $v_t$ is the difference between $x_t$ and $x_{t-1}$

$$\text{cosine similarity (vectors)} := \cos(\theta) = \frac{A \cdot B}{\|A\|\|B\|} \tag{5}$$

$$\text{cosine similarity (video frames)} := \cos(\theta) = \frac{x_t \cdot x_{t+1}}{\|x_t\|\|x_{t+1}\|} \tag{6}$$

$$\text{curvature} := c_t = \arccos(\hat{v}_t \cdot \hat{v}_{t+1}) \tag{7}$$

$$\text{cosine curvature} := \hat{v}_t \cdot \hat{v}_{t+1} = \frac{v_t \cdot v_{t+1}}{\|v_t\|\|v_{t+1}\|} = \cos(c_t) \tag{8}$$

# C  Models

## C.1  Model Sources

All Deep neural networks we analyzed were pretrained. The standard-trained ImageNet model was downloaded from PyTorch's model zoo [24]. Adversarially robust ResNet models were all downloaded from [13]. The adversarially robust ResNets were trained using projected gradient descent. All ViT [11] and standard trained CrossViT daggar [6] models were downloaded from the timm library [31]. All CrossViT daggar adversarially robust and rotationally invariant models were downloaded from the repository of Berrios and Deza [3]. The adversarially robust CrossViTs were trained with fast gradient sign method as stated in Berrios and Deza [3]. DINO models were downloaded from the DINO repository [5], while PredNet models were downloaded from Lotter et al. [22].

## C.2  Model Analysis Procedure

We showed each model the same 12 natural videos that were used in the psychophysics experiments of [18]. The videos were taken from the Chicago Motion Database [1], the film 'Dogville', Lions Gate Entertainment (2003), and LIVE Video Quality Database [29, 30]. The videos were grayscale, consisting of 11 frames each of $512 \times 512$ pixels, capturing natural motion such as rippling ocean water or a person walking through a crowded street. We resized the video frames to be $224 \times 224$ for all deep networks and $256 \times 256$ for bio-models that use steerable pyramids. One limitation of this work is that we did not evaluate models on larger video datasets, but we wanted to use psychophysical validated stimuli for our analyses. For each model, we recorded its activations at intermediate and

final layers for each video. We then found the global curvature for each stage of the model using equation 4 where we used the flattened model activations as the input $x_t$ to the curvature procedure.

We compared the global curvature at each layer of the model to the curvature of the video in pixel space. Models that straighten are defined as models that have a lower global curvature at deeper layers. When comparing the curvature of different model layers, we chose not to reduce the dimensionality of each layer activation to be the same across stages. Although principle components analysis (PCA) was sometimes used in Hénaff et al. [18] when expressing curvature, they did not use it in their analysis of deep networks. Furthermore, while an architecture's inherent dimensionality is likely relevant to a representation's curvature, we preferred not to introduce any additional transformations that would influence the measured curvature.

## C.3    Compute

Our methods do not require large compute. All individual model analyses can be run on CPU. We used a single GPU to speed up getting the features activations at each layer to the order of minutes per model.

## C.4    Adversarial Accuracy and Curvature

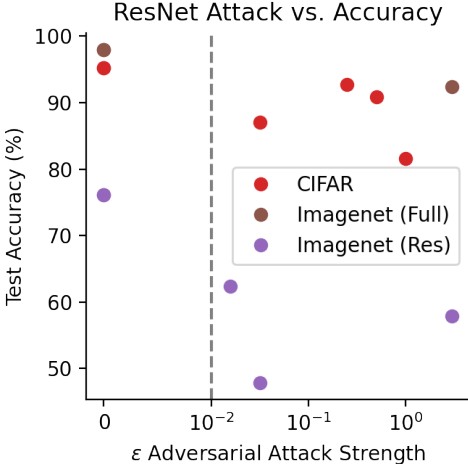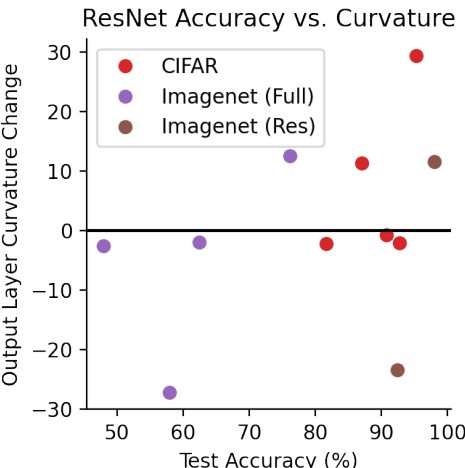

Figure 7: (Left): While adversarial attacks with greater  strength impart many desirable robustness properties on a network, adversarial training does not improve test accuracy, often decreasing test accuracy on the within-domain test set for a given model. Data plotted on symlog scale. (Right): While stronger adversarial attacks decrease curvature, improved test accuracy for a model is not predictive of output curvature reduction. Rather, within a given model training/test set, increased test accuracy predicts a smaller curvature reduction in the output layer.

## C.5    Negative Societal Impacts

We believe there are few negative societal impacts of this paper. Our work was exploratory and did not introduce any new models. The only negative impacts may be the general loss of jobs and industries that may result from artificial intelligence replacing human workers

