# OpenReview forum: "Exploring the perceptual straightness of adversarially robust and biologically-inspired visual representations"
_NeurIPS.cc/2022/Workshop/SVRHM — SVRHM Poster_

### Official Review · Reviewer_JVAo · 2022-10-14
**relevant for discussion although interpretation uncertain**

**Rating:** 6
**Confidence:** 3

**Review:**

This paper quantified the curvature of deep net model representations along time in natural videos. The authors identified a broad pattern: adversarially trained models had lower curvature in both their outputs and internal representation compared to the input (pixels) and non-adversarially trained model versions. Meanwhile, an equally adversarial robust, but not adversarially trained, model---VOneCORnet-S---did not reduce curvature at its output. VOneCORnet-S decreased curvature only in its biologically based portion, as did other models more closely matching biology.

I thought the results were sufficiently interesting and relevant for discussion in the workshop. Notwithstanding this, I thought the study design and interpretation were limited in many ways. Each limitation might be modest but together made it difficult to draw firm conclusions. I hope the authors may find my questions and suggestions useful.

- Conceptually, it seemed unreasonable to measure curvature along time for most of the tested models, which knew and cared nothing about time. I understand it was desirable to have data comparable to psychophysics data. However, commensurability matters little if it implies, for example, testing a Ping Pong RNN on King Kong. Incidentally, what were the units of curvature reported throughout the paper? Presumably, the units were degrees and comparable to, e.g., the x-axis values in Fig. 3c of Hernaff et al., 2019. But it would be better to state this correspondence explicitly.

- It seemed careless to compare curvature across representation spaces that might have vastly different dimensionality. Curvature is a function of the angle between vectors, and random vectors are more likely to be orthogonal in higher dimensional space. This might not be a significant issue if, for example, the representations tested were all high-dimensional. (Were they? What was the dimensionality of each representation you quantified?) Nevertheless, one cannot know without empirical verification. Pairwise comparisons were interpretable within the same model, adversarially trained or not, because both versions shared architecture and dimensionality. However, I could not be sure how valid the comparisons were across model architectures.

- The interpretation was only half spelled out as to why PredNet did not reduce curvature. The authors attributed this result to PredNet producing outputs in pixel space. That was certainly true, but probably more important was that PredNet predicted the next frame. An ideal PredNet would output the same video shifted in time, trivially leading to unchanged curvature.

- Why did the authors only test the CORnet version of VOneNet? CORnet was an unusual architecture in some sense. It would be informative to compare VOneCORnet to VOneNetResNet50 to see whether the results were due to the CORnet backend.

---

### Official Review · Reviewer_uNx2 · 2022-10-14
**Interesting work very relevant to the workshop**

**Rating:** 7
**Confidence:** 4

**Review:**

This paper explores the "straightness" of representations of video in the hidden layers of different kinds of networks.  They find that adversarial training leads to straighter representations, biologically inspired early processing increases straightness at those layers, but not necessarily in later layers.

Pros:
- This is interesting work that provides another way to compare different types of architectures and how well they do or don't correspond to human vision.
- It is very relevant to the workshop
- The removal of noise from the V1-block of VOneNetCornets was a nice experiment

Cons:
- The paper could be written a bit more clearly, though the page limitation is likely a source of some of the reduced clarity.
- Other insights into how adversarial training increases straightness would also make the paper stronger
- Other insights, for example, from examining what is happening at the VOneNetCornets V1-block, would make the paper stronger

---

### Official Review · Reviewer_nMKT · 2022-10-14
**The paper is well written, and the main results and conclusions are presented in a clear manner. Moreover, the use of a variety of networks makes the main claims more robust.**

**Rating:** 8
**Confidence:** 3

**Review:**

Quality
The quality of the project is good. Comparisons are made between a variety of network architectures, trained on a variety of tasks, which results in convincing conclusions and interpretations of the results.

Clarity
The order with which things are introduced made it a bit difficult for me to understand the motivation of the chosen training task (i.e. adversarial attacks) and the research metric (i.e. perceptual straightness). The paper starts with introducing the representational trajectories, followed by the distinction between biologically-realistic and unrealistic network architectures, ending with the introduction of training networks on adversarial robustness. Only later in the section Previous Work (pg. 2) it became clear to me what the link is between adversarial attacks and representational trajectories. Due to the fact that the motivation for training networks on adversarial attacks, and its relation to representational straightness only becomes evident in the end of the introduction, in the beginning the research objective comes across slightly nonintuitive. Briefly mentioning their relation in the beginning of the introduction might resolve this.

In the introduction it is mentioned that trajectories straighten as you go higher up in the visual hierarchy (which is also illustrated in Figure 1). It would maybe make a stronger case if this is backed up with a reference. The paper of Hénaff et al. [18] is mentioned, but this only states that compared to pixel space, human perceptual space is less curved. Also, the conclusion written in the last sentence before the paragraph of Previous Work, does in my opinion not really reflect what is shown in the result sections. That is, it seems that biologically-realistic networks show trajectories that are more similar to those in the brain?

It would maybe have made more sense to first show the results presented in section 7 (pg. 4) followed by section 6, since in the first figure you already show that biological networks show the least curved trajectory in the final output layer (which in the introduction you argue, is most similar to human representational spaces).

Lastly, while in the sectio Previous Work, the relationship between perceptual straightness and adversarial training is clearly explained, I had difficulty understanding the relevance of mentioning adversarial attacks in the context of transfer learning, since this is not examined or studied in the paper.

Originality
Though I'm not that familiar with representation trajectories (and what their role is in visual processing), I thought it is an interesting way of studying how the brain and neural networks represent the visual input over time. I was wondering what this metric says about the stability of representations over in the temporal space. Would a less straight curve mean that brain areas pertain to a more stable (less varying representation) over time?

Significance of this work
You could argue that the result showing biologically-realistic networks have a layer-wise curvature more similar to the human visual hierarchy is somewhat expected? I'm not sure what it would mean if you would have seen the opposite, namely that biologically-realistic network architectures have similar or even less straightened curvatures than biologically-unrealistic models. In other words, what does this finding tell you either about the brain or about neural networks?

Pros
Clear figures that reflect the results in an intuitive and coherent way. In addition, the use of a variety of networks and training tasks enforces the claims made.

Cons
The concept and definition of perceptual straightness is clearly explained, however, the implications of curviness in relation to how visual information is represented is not really explained, therefore making the metric a bit vague.

Other comments
Figure 1: as you go along the visual processing stream, the trajectories in the schematic seem to become shorter in addition to less curved, is this also somehow related to a visual processing as is performed in the brain?

Figure 2: would it be an idea to add error bars so that the plot becomes more informative and the conclusions drawn from the comparisons between networks are mor convincing?

Figure 3: Unclear to me what the individual points represent.

Figure 4: why are the adversarial counterparts only shown for resnet50 and crossvit?

---

### Official Review · Reviewer_kwz4 · 2022-10-16
**A well-written paper that offers insights for implementation-level vision models but lacks originality and has unclear implications for brain computation models at the algorithmic level.**

**Rating:** 6
**Confidence:** 3

**Review:**


# Summary

The study investigated representational straightness of natural video sequences across a large model set to explore what led to straighter representations. It demonstrated that adversarial training for different model architectures and non-parametric biological filter models showed straighter representation. Adding a biologically-inspired front end to neural networks did not guarantee downstream representational straightening.

# Quality / Clarity 6.5

The paper was in general well-written. The abstract and introduction clearly laid out that the objective of this study was to compare the effect of adversarial training, model architectures, and biologically inspired elements on representational straightness in models. However, the theoretical motivation behind straight representation was not demonstrated clearly. The paper discussed a normative account of why straighter representations might be useful (Henaff et al., 2019), but the role of representational straightness in early vision and perceptual space in our abilities to predict the future was still very hypothetical. Without a more compelling demonstration behind this normative hypothesis, model representational curvature should not be an important evaluation metric for vision models at the algorithmic level. The fact that low-level, non-parametric early vision filter models performed the best with the lowest output curvature also showed that this metric was limited, because these filter models could not perform any human vision task. Nonetheless, those who are interested in at the brain implementation level could definitely benefit from this computationally cheap evaluation metric and this paper’s demonstration on what makes a model learn straighter representations.

# Originality 5

The study was somewhat novel in that it compared the representational straightness across a large set of models, which included different architectures, model training method, and training objectives. It also showed detailed and careful comparison across model layers, the types and strengths of adversarial attacks, parametric and non-parametric biologically-inspired models. It was really interesting to see how adversarial training benefited models with very different architectures on straightening representations of video sequences.

However, the models were all pretrained, and the method and dataset for measuring global curvature in the models were both previously proposed in Henaff et al. (2019). Some interesting ideas in Henaff et al. (2019), such as the condition using naturalistic, intensity-linear image sequences, were not analyzed in the current study.

# Significance 6.5

The study offered insights for those working on the implementation models of vision. It showed that adversarially trained and biologically-inspired models have straighter representation, in alignment with early vision and perception. Therefore, it could be a good constraint and computationally cheap evaluation metric for implementation-level models. However, it lacked theoretical motivation on the functional importance of measuring representational straightness, so it is unclear if the metric is important for researchers interested in the computational mechanisms and algorithms of human vision. More theoretical demonstrations on the functional importance will be useful.

The study also engaged in the broader discussion on the non-equivalence of adversarial training and adversarial robustness and showed that VOneNet, despite being robust, did not have straighter representations in downstream layers, unlike adversarially trained models. It will be interesting to investigate what made adversarial training important for the straightening property to emerge.

---

### Official Review · Reviewer_31K9 · 2022-10-17
**The paper studies the curvature of different neural networks and presents interesting behaviours across different experiment choices, albeit with little information and no analysis why that might be the case.**

**Rating:** 7
**Confidence:** 4

**Review:**

## Clarity
The paper is well-written and easy to understand.

## Originality and Summary
The paper makes use of the curvature idea proposed by (Hénaff et al.), but applies it in a novel way, namely to neural network embeddings to understand the curvature of the artificial neural networks. This leads to several interesting insights: biologically-inspired neural networks and adversarial training lead to lower curvature. They compare several different models across different experimental choices: architecture type (biological, deep networks, forecasting networks, robust networks), training type (adversarial or not), and across layers.

## Quality
The experimental results are interesting. There are, however, many different behaviors not understood exactly why is that the case. For example, it can be seen across different architectures that biologically-inspired models have more straight representations in the embedding space across the different layers. Although this can be seen experimentally, there is no discussion of why that might be the case. Maybe the different architectural choices of the models could also be presented, with an analysis of the particular choices of the architectures and the training method that lead to straighter representations.  To put it short, although the experiments give interesting results, on a more detailed level, there is no understanding of which particular choices lead to these results. For example, is there some architectural building block that plays the most important role in the reduction of the curvature, etc. Extensions of this work call for the analysis of such questions.

## Significance
The problem analyzed in this paper is interesting. Using the curvature as a metric for the neural networks is interesting since it relates to many different experimental choices: biologically-inspired and adversarially trained networks have smaller curvature, whereas deep learning models trained non-adversarially trained models have not.